# Unusual Bicyclo[3.2.1]Octanoid Neolignans from Leaves of *Piper crocatum* and Their Effect on Pyruvate Dehydrogenase Activity

**DOI:** 10.3390/plants10091855

**Published:** 2021-09-07

**Authors:** Yu Juan Chai, Younghoon Go, Hai Qi Zhou, Hong Xu Li, Sun Joo Lee, Yeo Jin Park, Wahyu Widowatib, Rizal Rizal, Young Ho Kim, Seo Young Yang, Wei Li

**Affiliations:** 1Health Science Center, School of Biomedical Engineering, Shenzhen University, Shenzhen 518060, China; chaiyj@szu.edu.cn; 2Korean Medicine (KM) Application Center, Korea Institute of Oriental Medicine, Daegu 41062, Korea; gotra827@kiom.re.kr (Y.G.); pyjin5526@kiom.re.kr (Y.J.P.); 3Shenzhen Key Laboratory of Marine Bioresource and Eco-Environmental Science, College of Life Sciences and Oceanography, Shenzhen University, Shenzhen 518060, China; 2170257307@szu.edu.cn (H.Q.Z.); lhx@szu.edu.cn (H.X.L.); 4New Drug Development Center, Daegu-Gyeongbuk Medical Innovation Foundation, 80 Cheombok-ro, Dong-gu, Daegu 41061, Korea; disjrk@dgmif.re.kr; 5Korean Convergence Medicine, University of Science and Technology, Daejeon 34054, Korea; 6Medical Research Centre, Faculty of Medicine, Maranatha Christian University, Bandung 40164, Indonesia; wahyu_w60@yahoo.com; 7Biomolecular Biomedical Research Center for Biology, Indoensian Institute of Sciences, Bandung 40163, Indonesia; rizal_biotek@yahoo.com; 8Biomedical Engineering, Department of Electrical Engineering, Faculty of Engineering, Universitas Indonesia, West Java 16424, Indonesia; 9College of Pharmacy, Chungnam National University, Daejeon 34134, Korea; yhk@cnu.ac.kr; 10Department of Pharmaceutical Engineering, Sangji University, 83 Sangjidae-gil, Wonju-si 26339, Korea

**Keywords:** *Piper crocatum* Ruiz & Pav, Piperaceae, antineoplastic cell agents, neolignan

## Abstract

Three undescribed bicyclo[3.2.1]octanoid neolignan glucosides, along with a known neolignan, were isolated from the leaves of *Piper crocatum* Ruiz & Pav. Their chemical structures were elucidated using extensive spectroscopic analyses including 1D and 2D NMR experiments and HR-ESI-MS analysis, as well as through comparison with previously reported data. Two compounds were assessed for their inhibitory effect against pyruvate dehydrogenase E1α S300 phosphorylation. The fluorescent image suggested that both compounds (60 µM) revealed a stronger inhibition effect than the positive control (dichloroacetate, DCA 5 mM), with IC_50_ values of 99.82 µM and 80.25 µM, respectively.

## 1. Introduction

The genus *Piper* belongs to the Piperaceae family, considered as one of the largest genera among angiosperms and one of the most abundant species in tropical forests [1]. Several species have high economic value, such as *Piper nigrum* (black pepper), the representative species of the Piperaceae family used worldwide as a condiment having medicinal properties. *Piper* species have been widely investigated, and a number of physiological activity components have been isolated, which can be characterized into the typical classes of alkaloids, amides, chalcones, dihydrochalcones, flavanones, lignans, neolignans, propenylphenols, steroids, and terpenes [1]. However, only a few chemical investigations of *P. crocatum* have been reported, which were limited to sterols, lignans, and neolignans [2,3,4]. In China, people have used the dried stem of *Piper kadsura* (Choisy) Ohwi as a traditional Chinese medicine to treat asthma and rheumatism for hundreds of years. The species of *P. crocatum* is commonly used in folk medicine in Indonesia to treat various diseases [2]. Many species of the Piperaceae family have been extensively studied for their potential antitumor, antimicrobial, and antifungal properties [2,5]. Therefore, the chemical constitution of *P. crocatum* still needs to be investigated because it is a source of novel natural products with potential activities. The metabolic characteristics of neoplastic and non-neoplastic cells are considerably different. However, non-neoplastic cells predominantly depend on ATP/energy produced by pyruvate oxidation in the mitochondria, and each molecular glucose oxidized generates 36 ATPs; whereas proliferating cancer cells predominantly rely on aerobic glycolysis in the cytoplasm, and up to four ATPs are produced for each glucose molecule [6]. Pyruvate dehydrogenase complex and pyruvate dehydrogenase kinase (PDK) are key mitochondrial enzymes in the metabolic pathway of glucose, and their interaction regulates the proportion between aerobic respiration and the Warburg effect [7,8]. Because all tumors rely on metabolic alterations for growth, metastasis, and survival, atypical pathways may be potential targets of antineoplastic drugs.

## 2. Results

### Structural Elucidation 

In this study, four bicyclo[3.2.1]octanoid neolignans were isolated from the MeOH extract of *P. crocatum* leaves. Their structures were identified as pipcroside A (**1**), pipcroside B (**2**), pipcroside C (**3**), and crocatin B (**4**) [3]. Their structures were elucidated by one- and two-dimensional NMR and mass spectrometries (Figure 1) (see Appendix A).

To our knowledge, this study is the first to isolate bicyclo[3.2.1]neolignan glucoside derivative compounds from the natural product and macrophyllin-type neolignans from *P. crocatum*.

## 3. Discussion

Compound **1** was isolated as a white amorphous powder. Its molecular formula was established as C_27_H_36_O_11_ using HR-ESI-MS. The ^1^H NMR spectrum (Table 1) showed the presence of an ABX system of aromatic protons at δ_H_ 6.47 (t, J = 2.0 Hz, H-2), 6.48 (dd, J = 8.0, 2.0 Hz, H-4), and 6.87 (d, J = 8.0 Hz, H-5), as well as an allyl group suggested from the COSY spectrum at δ_H_ 2.30 (dd, J = 13.7, 7.0 Hz, H-7′a), 2.63 (dd, J = 13.7, 7.0 Hz, H-7′b), 5.14 (td, J = 10.0 Hz, H-9′a), 5.27 (d, J = 17.0 Hz, H-9′b), and 5.93 (td, J = 17.0, 7.0 Hz, H-8′), and 3.24 (m, H-7)/2.13 (p, J = 7.0 Hz, H-8), and a signal assigned to the anomeric proton of the glucosyl moiety at δ_H_ 4.84 (d, J = 7.2 Hz, H-1″). Other proton signals suggested three methoxyl groups at δ_H_ 3.16 (s, 3′-OMe), 3.56 (s, 5′-OMe), and 3.64 (s, 6-OMe), as well as a methyl group attached to the aliphatic carbon at δ_H_ 1.15 (d, J = 7.0 Hz, H-9). The ^13^C NMR spectrum indicated the presence of six aromatic carbons at δ_C_ 112.4 (C-2), 114.3 (C-5), 121.4 (C-4), 132.4 (C-1), 145.2 (C-3), and 148.0 (C-6) and a glucosyl moiety at δ_C_ 60.4 (C-6″), 69.5 (C-4″), 73.1 (C-2″), 76.8 (C-3″), 76.9 (C-5″), and 99.6 (C-1″). Other signals, such as a carbinol carbon at δ_C_ 78.1 (C-2′); an allyl group at δ_C_ 34.1 (C-7′), 118.0 (C-9′), and 135.3 (C-8′); olefinic carbons at δ_C_ 126.6 (C-6′) and 151.6 (C-5′); two quaternary carbons at δ_C_ 48.6 (C-1′) and 96.1 (C-3′); and a carbonyl group at δ_C_ 192.4 (C-4′), indicated a bicyclo[3.2.1]octane derivative, which was similar to compound **4**. Therefore, both ^1^H and ^13^C NMR spectra demonstrated that compound **1** was a bicyclo[3.2.1]octane lignan derivative. The connection between aromatic ring, glycosyl group, and bicyclo[3.2.1]octane of compound **1** was unambiguously assigned based on HMBC spectra (Figure 2). Key HMBC correlations between H-2′ (δ_H_ 3.81)/C-4′ (δ_C_ 192.4) and C-7 (δ_C_ 57.9) indicate the connection between the C_3_-bicyclooctane moiety at C-3′. Similarly, H-7′ (δ_H_ 2.30 and 2.63)/C-1′ (δ_C_ 48.6), C-2′ (δ_C_ 78.1), C-6′ (δ_C_ 126.6), and C-8 (δ_C_ 47.8) further supported the presence of the C_3_-bicyclooctane moiety at C-1′ (δ_C_ 48.6) and the assigned position of an allyl group at C-1′. The correlation between H-1″ (δ_H_ 4.84) and 145.2 (C-3) indicated that the glycosyl group was attached at C-7 of the aromatic ring. These data led to the identification of a guianin-type 7.1′,8.3′-connected neolignan skeleton of compound **1**.

The relative stereochemistry of methyl (H-9) and aryl groups was revealed from the key ROESY correlation between H-9 and H-7, and no correlation was observed between H-9 and aromatic ortho-protons (H-2 and H-6). However, a correlation between H-8 and H-4 was observed. Therefore, the trans-relationship of methyl (H-9) and aryl groups was confirmed [4]. Additionally, the correlations between H-2′, –OH-2′, and H-9 were observed, which indicated the endo conformation of the hydroxy group on C-2′. The correlation between H-9 and –OH-2′, which indicated the relative configuration of the hydroxy group, suggested endo configuration with the methyl (H-9) group. The relative configuration of **1** was further determined based on the reported chemical shifts of H-7, H-6′, and methyl at C-8. In 1′S, 2′S, 3′R configuration series, H-7 was determined to be resonated at δ_H_ 2.6, H-6′ at δ_H_ 5.7, and methyl between δ_H_ 0.8 and 0.9, whereas in 1′R, 2′R, 3′S configuration series, the resonances for H-7, H-6′, and methyl protons appeared at δ_H_ 3.3, 6.2, and 1.25–1.28, respectively [9]. For our study, the chemical shifts of H-7, H-6′, and H-8 methyl were δ_H_ 3.24, 6.29, and 1.15, respectively. The absolute configuration of compound **1** was deduced to be (1′,2′R,3′S,7S,8R)-neolignan, and two adjacent torsion angles were measured (C2′–C3′–C4′–C5′ = −37.7° and C2′–C1′–C6′–C5′ = 36.5°) and compared with those reported in the literature [10] (see Appendix A). Based on the abovementioned data, the structure of compound **1** was determined as (1′R,2′R,3′S,7S,8R)-∆^5^′^,8^′-2′-hydroxy-6,3′,5′-trimethoxyl-4′-oxo-8.1′7.3′-neolignan-3-*O*-β-d-glucopyranoside, and the compound was named pipcroside A.

Compound **2** was isolated as a colorless gum. The molecular formula was established as C_28_H_38_O_12_ using HR-ESI-MS. The ^1^H and ^13^C spectroscopic data (Table 1) of compound **2** were determined to be identical to those of compound **4** except for that of the glucose moiety, and the signal was assigned to the anomeric proton of the glucosyl group at δ_H_ 4.66 (d, J = 7.2 Hz, H-1″). Thus, the absolute configuration of the C_3_-bicyclooctane moiety of compound **2** was the same as that of compound **4**, and the absolute configurations were confirmed using X-ray [3]. Based on the abovementioned data, the absolute configuration of compound **2** was determined as (1′R,2′R,3′S,7S,8R)-∆^5^′^,8^′-2′-hydroxy-3,5,3′,5′-tetramethoxyl-4′-oxo-8.1′7.3′-neolignan-4-*O*-β-d-glucopyranoside, and the compound was named pipcroside B.

Compound **3** was isolated as a colorless gum. The molecular formula was established as C_28_H_38_O_12_ using HR-ESI-MS. The ^1^H NMR spectrum data (Table 1) indicated the existence of four methoxyl groups, including two symmetric methoxyl groups substituted on the aromatic ring at δ_H_ 3.79 (s, 3-OMe and 5-OMe), two methoxyl groups attached to aliphatic carbons at δ_H_ 3.34 (s, 1′-OMe) and 3.49 (s, 3′-OMe), and an allyl group at δ_H_ 5.86 (dd, J = 17.0, 10.0 Hz, H-8′), 5.11 (ddd, J = 17.0, 3.0, 2.0 Hz, H-9′), 2.92 (ddd, J = 16.0, 7.0, 1.0 Hz, H-7′), and 2.99 (ddd, J = 16.0, 7.0, 1.0 Hz, H-7′). The ^1^H NMR chemical shift of the C-8 methyl group at δ_H_ 1.28 (d, J = 7.0 Hz, H-9) and other compounds indicated a trans-relationship between the C-7 aryl and C-8 methyl groups. However, the ^13^C NMR spectrum showed one quaternary carbon δ_C_ 86.2 (C-1′) along with olefinic carbons at δ_C_ 139.5 (C-5′) and 156.5 (C-6′), which were considerably different from the other three compounds. The HMBC correlations further confirmed these connections among the bicyclooctane moiety through colo1′-OMe (δ_H_ 3.34)/C-1′ (δ_C_ 86.2), which indicated the deshielding effect of C-1′ (δ_C_ 86.2) compared with the other three compounds. Together with H-7′ (δ_H_ 2.92 and 2.99), C-5′ (δ_C_ 139.5), C-6′ (δ_C_ 156.5), and C-4′ (δ_C_ 199.1) confirmed that the allyl group was connected to an olefinic carbon C-5′ (δ_C_ 139.5). Subsequently, by comparing with the reported literature [6,11], compound **3** was considered to comprise a macrophyllin-type bicyclo[3.2.1]octanoid neolignan glucoside compound, and two adjacent torsion angles were measured (C2′–C3′–C4′–C5′ = 39.3° and C2′–C1′–C6′–C5′ = −34.8°) and compared with the reported literature (Coy-Barrera et al. 2012) (see Appendix A). Hence, the relative configuration for the C_3_-bicyclooctane moiety of compound **3** was determined to be (1′R,2′S,3′R,7S,8R)-∆^5^′^8^′-2′-hydroxy-3,5,1′,3′-tetramethoxyl-4′-oxo-8.1′7.3′-neolignan-4-*O*-β-d-glucopyranoside, and the compound was named pipcroside C.

Reports from the literature were compared [12,13,14] for the total synthesis of bicyclo[3.2.1]octanoid neolignans. The speculated biosynthetic pathways are in the Appendix A.

### PDH Inhibition Effect

PDH is essential in the pathway of glucose metabolism and ATP production. Suppression of PDH reduces the aerobic oxidation of pyruvate and promotes its transformation to lactate in cytoplasm. Additionally, decreased flux of pyruvate via PDH diminishes the amount of acetyl-CoA entering into the TCA cycle. Accordingly, defects in both PDH and oxidative phosphorylation activities result in lactic acidosis and initiation of the Warburg effect [7]. The activity of PDH is mainly controlled by reversible phosphorylation of the E1α subunit on residue sites serine 293 (Site 1), 300 (Site 2), and 232 (Site 3), which are mediated by PDK and pyruvate dehydrogenase phosphatase (PDP) phosphorylation of E1α by PDKs, which inactivates PDH, whereas PDP exhibits active/reactive catalytic activity by dephosphorylating the complex [15]. Compounds **2** and **3** were evaluated for their inhibitory effect against PDHE1α S300 phosphorylation; we discuss the phosphorylation inhibitory effect of the E1α subunit on residue site serine 300 (Site 2). Fluorescent image results suggested that compounds **2** and **3** inhibited phosphorylation of E1α S300 at 60 µM with better effect than that of dichloroacetate (DCA) at 5 mM concentration; compound **2** showed the strongest effect among all three compounds (Figure 3).

The IC_50_ values of compounds **2** and **3** were rather high; that is, 99.82 and 80.25 µM, respectively. For the positive control DCA, the IC_50_ values were 1.03 µM with 0.1%, 0.97 µM with 0.6%, and 1.06 µM with 1.0% concentration, respectively (Figure 4).

Although previous research on DCA was promising, its application to anticancer treatment is limited by its non-specificity, low potency, and requirement for high doses to exhibit therapeutic effects, which always leads to peripheral neurological toxicity. The position exchange of the allyl group on C-1′ and methoxyl group on C-5′ resulted in a higher IC_50_ value difference between compounds **2** and **3** and revealed a better inhibitory effect of compound **2** than that of compound **3** and DCA in fluorescent imaging. Although compounds **2** and **3** possessed a rather high toxicity, the consumption of these two compounds was 83 times lower than that of DCA. The described type of neolignan compounds is one of the major components of *Piper* plants.

## 4. Materials and Methods

### 4.1. General Experimental Procedures

Optical rotations were determined using a Jasco DIP-370 automatic polarimeter (Jasco, Tokyo, Japan). The NMR spectra were recorded using a BRUKER AVANCE III 600 (^1^H, 600 MHz; ^13^C, 150 MHz) (Bruker Biospin GmbH, Karlsruhe, Germany), with tetramethylsilane (TMS) as an internal standard. Heteronuclear multiple quantum correlation (HMQC), heteronuclear multiple bond correlation (HMBC), rotating frame nuclear overhauser effect spectroscopy (ROESY), and ^1^H–^1^H correlation spectroscopy (COSY) spectra were recorded using a pulsed field gradient. The HR-ESI-MS spectra were obtained by using an Aglient 1200 LC-MSD Trap spectrometer (Agilent, Santa Clara, CA, USA). Preparative HPLC was performed using a GILSON 321 pump, 151 UV–VIS detector (Gilson, VILLIERS-LE-BEL, France), and RStech HECTOR-M C_18_ column (5-micron, 250 × 21.2 mm) (RS Tech Crop, Chungju, South Korea). Column chromatography was performed using a silica gel (Kieselgel 60, 70–230, and 230–400 mesh, Merck, Darmstadt, Germany), YMC C-18 resins, and thin layer chromatography (TLC) was performed using pre-coated silica-gel 60 F_254_ and RP-18 F_254S_ plates (both 0.25 mm, Merck, Darmstadt, Germany). The spots were detected under UV light and using 10% H_2_SO_4_. The adjacent torsion angles were measured by Chemdraw 3D (version 17.1).

### 4.2. Plant Material

Dried leaves of *P. crocatum* were collected from Cilendek Timur, Bogor, West Java, Indonesia, in August 2016 and taxonomically identified (Identification number: 1714/IPH.1.01/If.07/VIII/2016) by the staff at the herbarium Laboratory, Research Center for Biology, Indonesian Institute of Sciences, Cibinong, West Java, Indonesia. A voucher specimen (BBRC-SMPLS-003) was deposited at the Herbarium of Aretha Medika Utama, Biomolecular and Biomedical Research Center, Bandung, West Java, Indonesia.

### 4.3. Extraction and Isolation

The dried leaves of *P. crocatum* (2.6 kg) were reflux extracted with MeOH (5 L × 3 times). The total extraction (400.0 g) of MeOH was suspended in deionized water and partitioned with hexane, and water fraction. Then the water fraction was partitioned sequentially with EtOAc and BuOH, yielding EtOAc (1A, 16.1 g), BuOH (1B, 65.0 g). The EtOAc fraction was subjected to a silica gel column chromatography with a gradient of CHCl_3_-MeOH-H_2_O (10:1:0, 9:1:0, 8:1:0, 6:1:0.1, 5:1:0.1, 4:1:0.1, 3:1:0.1, 2:1:0.1, MeOH 2.0 L for each step) to give 11 fractions (Fr. 1A-1-1A-11). Fractions 5 and 6 were combined and isolated with a gradient of MeOH–water (1:2, 1:1, and MeOH) by MPLC using a YMC C_18_ column to give 8 fractions (Fr. 2A-1-2A-8). Subfraction 2A-8 was isolated by prep-HPLC (MeOH: H_2_O = 60:40 retention time = 41.6 min) to give compound **4** (100.0 mg). The BuOH fraction was subjected to silica-gel column chromatography with a gradient of CHCl_3_-MeOH-H_2_O (10:1:0, 9:1:0, 8:1:0, 6:1:0.1, 5:1:0.1, 4:1:0.1, 3:1:0.1, 2:1:0.1, MeOH 5.0 L for each step) to give 11 fractions (Fr. 2B-1-2B-11). Fractions 2B-2 and 2B-3 were combined and isolated with a gradient of MeOH–H_2_O (1:2, 1:1, and MeOH) by MPLC using a YMC C_18_ column to give 12 fractions (Fr. 1C-1-1C-12). The fraction 1C-10 was separated by a Sephadex LH-20 column and eluted by MeOH and its subfraction was isolated by prep-HPLC (MeOH: H_2_O = 45:55 retention time = 55.1 min and then ACN: H_2_O = 20:80 retention time = 62.3 min) to give compound **3** (4.0 mg). The fraction 1C-12 was separated by a Sephadex LH-20 column and eluted by MeOH and its subfraction was isolated by prep-HPLC (MeOH: H_2_O = 80:20 retention time = 95.2 min) to give compound **1** (5.0 mg). Fractions 2B-4 and 2B-5 were combined and isolated with a gradient of MeOH–H_2_O (1:2, 1:1, and MeOH) by MPLC using a YMC C_18_ column to give 13 fractions (1D-1-1D-13). The fraction 1D-13 was separated by a Sephadex LH-20 column and eluted by MeOH, and its subfraction was isolated by prep-HPLC (MeOH: H_2_O = 50:50 retention time = 44.5 min) to give compound **2** (60.0 mg). The percentage of the undescribed compounds in the extract plant were: 12.5 ppm (compound **1**), 150.0 ppm (compound **2**), and 10.0 ppm (compound **3**), respectively. The percentage of the undescribed compounds in the dry plant were: 1.9 ppm (compound **1**), 23.1 ppm (compound **2**), and 1.5 ppm (compound **3**), respectively.

### 4.4. Pyruvate Dehydrogenase Complex (PDH) Cellular Activity

Point two percent gelatin was added to a black 96-well plate with a clear bottom and incubated for 1 h. Afterwards, the plate was washed with growth media. Human AD-293 cells, derived from the HEK293 cells, were seeded into black 96-well plates with a clear bottom and grown for 24 h. Compounds were then added and incubated for 24 h. The cells were then fixed with 2% paraformaldehyde, and permeabilized anti-PDHE1 pSer300 (Merk Millipore, Burlington, MA, AP1064) was added and incubated overnight. Next, the cells were washed and Alexa fluor 488, goat anti-rabbit ab (Invitrogen, Waltham, MA, A11008) was added with Hoechst 33,258 (Invitrogen, H3569) and incubated for 2 h. Finally, cells were washed, and the plates were measured in Operetta (PerkinElmer, Waltham, MA). The raw data were normalized for the pharmacological inhibitor control (5 mM dichloroacetate (DCA)) and percentage effect values using the software package Harmony High-Content Imaging and Analysis Software 3.1. Dose response curves were generated by plotting the percentage effect values and IC_50_ values were calculated via GraphPad Prism 6.

### 4.5. Statistical Analysis

All values are expressed as means ± standard error of the mean. The statistical significance threshold (*p* < 0.05 for all analyses) was assessed by one-way ANOVA followed by Tukey’s post hoc test for multiple comparisons using Prism 5.01 software (GraphPad Software Inc., San Diego, CA, USA).

### 4.6. Spectroscopic Data

Compound **1**. White amorphous powder; ${\left[\alpha \right]}^{\overset{20}{\underset{D}{}}}$: +31.2 (c 0.051, MeOH); ^1^H NMR (dimethyl sulfoxide-*d_6_*, 600 MHz,) and ^13^C NMR data (dimethyl sulfoxide-*d_6_*, 150 MHz), see Table 1; HR-EI-MS *m*/*z*: 559.2159 [M + Na]^+^ (calcd for C_27_H_36_O_11_Na, 559.2155).

Compound **2**. Colourless gum; ${\left[\alpha \right]}^{\overset{20}{\underset{D}{}}}$: +43.04 (c 0.049, MeOH); ^1^H NMR (acetone-*d_6_*, 600 MHz,) and ^13^C NMR (acetone-*d_6_*, 150 MHz), see Table 1; HR-EI-MS *m*/*z*: 589.2264 [M + Na]^+^ (calcd for C_28_H_38_O_12_Na, 589.2261).

Compound **3**. Colourless gum; ${\left[\alpha \right]}^{\overset{20}{\underset{D}{}}}$: −32.64 (c 0.053, MeOH); ^1^H NMR (methanol-*d_4_*, 600 MHz,) and ^13^C NMR (methanol-*d_4_*, 150 MHz), see Table 1; HR-EI-MS *m*/*z*: 589.2275 [M + Na]^+^ (calcd for C_28_H_38_O_12_Na, 589.2261).

## 5. Conclusions

The bicyclo[3.2.1]octanoid neolignans constitute a significant category of neolignan compounds, which were classified as the guianin- and macrophyllin-type. Among the known bicyclo[3.2.1]octanoid neolignans, the later type neolignans are less common than the former one. Despite being the less common type of neolignan, the glucoside derivative compound of the macrophyllin-type was isolated. Along with the other glucoside derivative neolignans, they were for the first time reported from natural products. Meanwhile, biosynthetic pathways were speculated in comparison with total synthesis. To the best of our knowledge, this is the first study on the phosphorylation inhibitory effect of bicyclo[3.2.1]octanoid neolignan compounds. Additionally, this study provides a basis for further exploration of *P. crocatum* and bicyclo[3.2.1]octanoid neolignans from *Piper* plants as a new source of natural antineoplastic agents.

## Figures and Tables

**Figure 1 plants-10-01855-f001:**
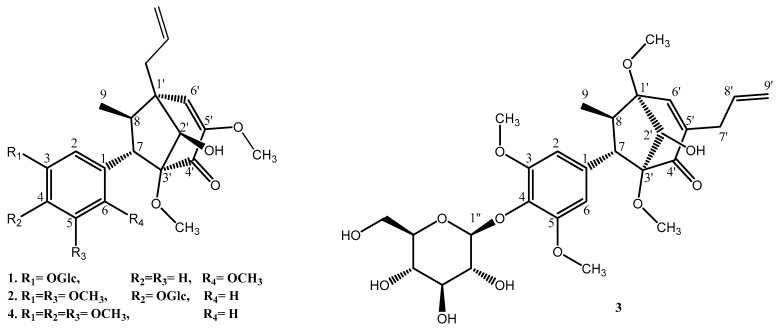
Structures of compounds **1**–**4** from *P. crocatum*.

**Figure 2 plants-10-01855-f002:**
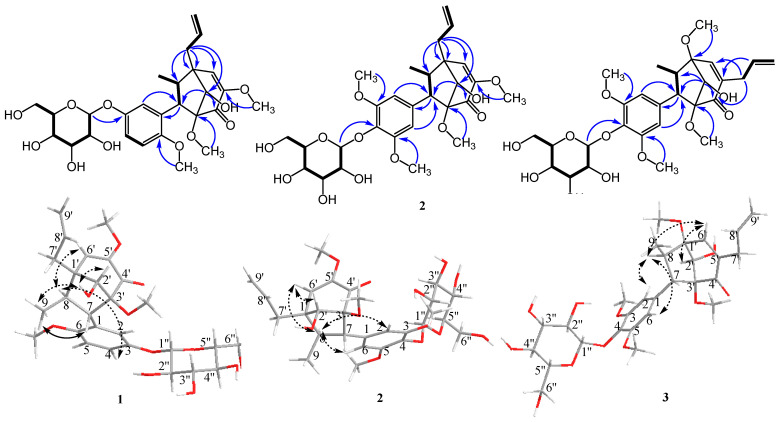
Key COSY (bold lines), HMBC (H→C), and ROESY (dashed arrows) correlations for new compounds (**1**–**3**).

**Figure 3 plants-10-01855-f003:**
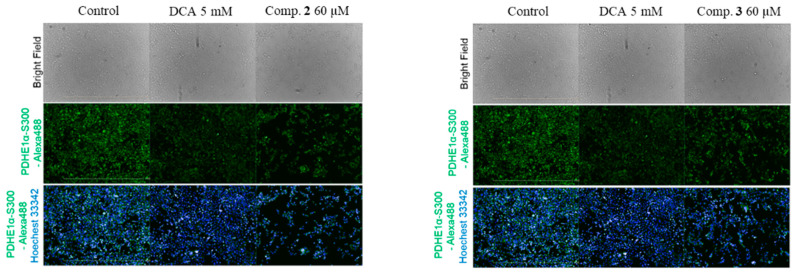
Fluorescent images of compounds **2** and **3**. Immunofluorescence analysis of p-PDH E1α (Ser300) in AD-293 cells (green) treated with dichloroacetic acid (DCA) and compounds isolated from *P. crocatum*. The cell nuclei (blue) were stained with Hoechst 3334.

**Figure 4 plants-10-01855-f004:**
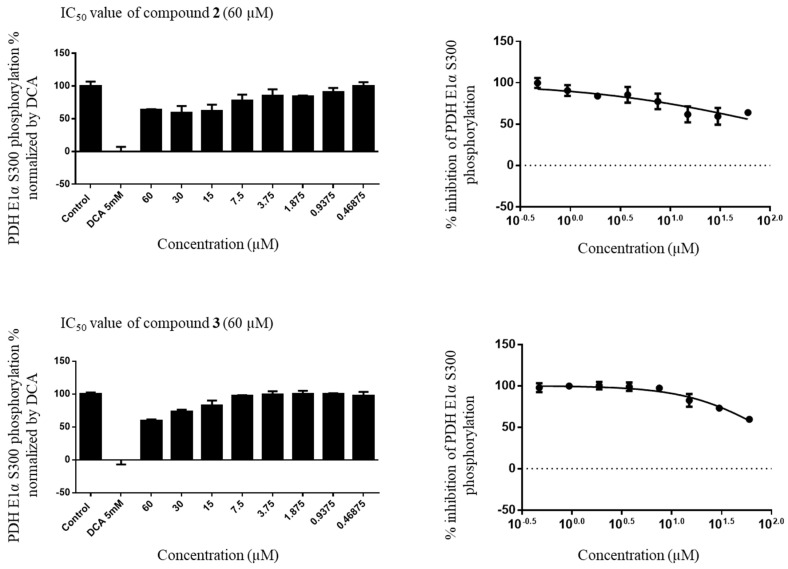
IC_50_ value of compounds **2** and **3**.

**Table 1 plants-10-01855-t001:** ^1^H and ^13^C NMR data of compounds **1**–**3**.

	Pipcroside A (1) ^a^		Pipcroside B (2) ^b^		Pipcroside C (3) ^c^	
Position	δ_C_ ^d^	δ_H, mult (J, Hz)_ ^e^	δ_C_ ^d^	δ_H, mult (J, Hz)_ ^e^	δ_C_ ^d^	δ_H, mult (J, Hz)_ ^e^
1	132.4		136.1		135.8	
2	112.4	6.47, t (2.0)	107.9	6.40, s	108.1	6.34, s
3	145.2		153.1		153.7	
4	121.4	6.48, dd (8.0, 2.0)	135.3		135.5	
5	114.3	6.87, d (8.0)	153.1		153.7	
6	148.0		107.9	6.40, s	108.1	6.34, s
7	57.9	3.24, m	60.0	3.38, d (7.0)	58.8	3.48, d (7.0)
8	47.8	2.13, p (7.0)	49.3	2.29, p (7.0)	48.0	2.38, q (7.0)
9	17.0	1.15, d (7.0)	17.6	1.27, d (7.0)	16.5	1.28, d (7.0)
1′	48.6		49.7		86.2	
2′	78.1	3.81, d (4.7)	79.4	4.04, s	77.1	4.35, s
3′	96.1		97.2		96.5	
4′	192.4		192.7		199.1	
5′	151.6		153.3		139.5	
6′	126.6	6.29, s	126.4	6.31, s	156.5	7.36, s
7′a	34.1	2.30, dd (13.7, 7.0)	35.2	2.39, dd (13.7, 7.0)	33.2	2.92, ddd (16.0, 7.0, 1.0)
7′b		2.63, dd (13.7, 7.0)		2.79, dd (13.7, 7.0)		2.99, ddd (16.0, 7.0, 1.0)
8′	135.3	5.93, td (17.0, 7.0)	136.1	5.98, ddd (17.0, 10.0, 7.2)	135.8	5.86, dd (17.0, 10.0)
9′a	118.0	5.14, d (10.0)	118.2	5.13, dd (10.0, 1.7)	117.7	5.11, ddd (17.0, 3.0, 2.0)
9′b		5.27, d (17.0)		5.28, dd (17.0, 1.7)		
3-OMe			56.6	3.75, s	57.0	3.79, s
4-OMe						
5-OMe			56.6	3.75, s	57.0	3.79, s
6-OMe	55.1	3.64, s				
1′-OMe					54.3	3.34, s
3′-OMe	53.1	3.16, s	54.1	3.30, s	53.6	3.49, s
5′-OMe	54.8	3.56, s	55.3	3.65, s		
1′′	99.6	4.84, d (7.2)	106.1	4.66, d (7.2)	105.3	4.80, d (7.5)
2′′	73.1	3.24, m	75.6	3.40, m	75.0	3.44, d (7.6)
3′′	76.8	3.24, m	77.6	3.43, m	77.8	3.41, m
4′′	69.5	3.16, m	71.4	3.43, m	71.2	3.41, m
5′′	76.9	3.24, m	77.6	3.27, dd (8.1, 4.4)	78.2	3.18, ddt (7.3, 5.0, 2.3)
6′′a	60.4	3.43, dd (11.7, 5.8)	62.3	3.66, m	62.5	3.67, dd (12.0, 5.0)
6′′b		3.66, m		3.77, dd (11.6, 3.3)		3.77, dd (12.0, 2.5)

Assignments were made based on HMQC and HMBC experiments; *J* values (Hz) are shown in parentheses. ^a^ Measured in dimethyl sulfoxide-*d_6_*; ^b^ measured in acetone-*d_6_*; ^c^ measured in methanol-*d_4_*; ^d^ 150 MHz; ^e^ 600 MHz. t: triplet; s: singlet; d: doublet; dd: double doublet; ddd: double double doublet; ddt: double double triplet; td: triple doublet; q: quartet; p: pentet; m: multiplet.

## Data Availability

Data is contained within the article or Appendix A.

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
