# Peer review of "Unusual Bicyclo[3.2.1]Octanoid Neolignans from Leaves of Piper crocatum and Their Effect on Pyruvate Dehydrogenase Activity"

_plants, 2021, doi:10.3390/plants10091855_

Round 1

Reviewer 1 Report

The structure elucidation of the new compounds is sound. However, some revision may improve the manuscript quality and scientific rigor.

-Pag 3, line 92: "The sequence between.." better " The connection between"

-Pag 3, line 106-107: Ortho-protons to the aryl ring are H2 and H6

-Pag 3, line 109-110:"The ROESY correlation of the H-8 and H-6' indicated..the enone conformation of the methyl group"? This sentence is unclear..

-Pag 3, line 110-11: "Additionally, the ROESY correlation between H-2′ and –OH-2′ was observed, which indicated the endo conformation of the hydroxy group on C-2′". This sentence it is not self-explanatory and should be explained.

-Attribution of absolute configuration of compounds 1-3: The 3-D structure of compound 1-3 has only been inferred by ROESY NMR, which can only provide information on the groups which show spatial proximity in the molecular structure. Thus, for a ring with limited conformational flexibility through ROESY NMR can only be found out which group are cofacial oriented. No information about absolute configuration can be inferred by ROESY. Similarly, from the dihedral angle and J coupling analysis only the relative configuration can be inferred. For the assignment of the absolute configuration, the authors refer to Lit ref 8-10, where the absolute configuration has been inferred through CD spectroscopy and X-Ray diffraction. With the data provided by the authors only the relative configuration of the new compounds can be inferred. The absolute configuration could only be assumed to be the same as for the previously isolated compounds, on the basis of biogenetic considerations. This point needs to be better discussed and clarified in the main text.

-Absolute configuration of sugar moieties: As explained above, by NMR (J coupling analysis and 13C shift) only the relative configuration of the sugar can be inferred. The assignment of D and L sugar can only be done by X-ray, or after hydrolysis of the sugar moiety followed by derivation to a diastereoisomer and comparison with the corresponding D and L standards by GC or HPLC. As above, by only NMR data, the D and L absolute stereochemistry of sugar can only be hypothesized based on the same biogenetic pathway of the compounds previously isolated for which abs configuration has been determined.

Author Response

Answers of Reviewer 1' Comments

Manuscript ID: plants-1365581
Title: Unusual Bicyclo [3.2.1] Octanoid Neolignans from Leaves of Piper crocatum and Their Effect on Pyruvate Dehydrogenase Activity

Corresponding Author: Dr. Wei Li

We greatly appreciate your kind consideration on our manuscript.

According to reviewers’ comments, several parts of the manuscript have been corrected.

All comments were answered one by one and highlighted by yellow colour.

Reviewer 1' comments to Author:

PC-1 plants

Comments and Suggestions for Authors

The structure elucidation of the new compounds is sound. However, some revision may improve the manuscript quality and scientific rigor.

-Pag 3, line 94: "The sequence between." better " The connection between"

Answer: This comment was corrected in the revised manuscript.

-Pag 3, line 109: Ortho-protons to the aryl ring are H2 and H6

Answer: This comment was corrected in the revised manuscript.

-Pag 3, line 111-112:"The ROESY correlation of the H-8 and H-6' indicated. The enone conformation of the methyl group"? This sentence is unclear.

Answer: The relative configuration of 1 was determined based on the reported chemical shifts of H-7, H-6′, and methyl at C-8. In 1′S, 2′S, 3′R configuration series, H-7 was determined to be resonated at δH 2.6, H-6′ at δH 5.7, and methyl between δH 0.8 and 0.9, whereas in 1′R, 2′R, 3′S configuration series, the resonances for H-7, H-6′, and methyl protons appeared at δH 3.3, 6.2, and 1.25–1.28, respectively. (Oscar, C.C.; Janzen, D.H.; Dodson, C.D.; Stermitz, F.R. Neolignans from fruits of Ocotea veraguensis. Phytochemistry 1987, 26, 2037-2040.) Therefore, we considered this sentence was irrelevant, and this sentence was delated.

-Pag 3, line 112-114: "Additionally, the ROESY correlation between H-2′ and –OH-2′ was observed, which indicated the endo conformation of the hydroxy group on C-2′". This sentence it is not self-explanatory and should be explained.

Answer: This comment was corrected in the revised manuscript.

-Attribution of absolute configuration of compounds 1-3: The 3-D structure of compound 1-3 has only been inferred by ROESY NMR, which can only provide information on the groups which show spatial proximity in the molecular structure. Thus, for a ring with limited conformational flexibility through ROESY NMR can only be found out which group are cofacial oriented. No information about absolute configuration can be inferred by ROESY. Similarly, from the dihedral angle and J coupling analysis only the relative configuration can be inferred. For the assignment of the absolute configuration, the authors refer to Lit ref 8-10, where the absolute configuration has been inferred through CD spectroscopy and X-Ray diffraction. With the data provided by the authors only the relative configuration of the new compounds can be inferred. The absolute configuration could only be assumed to be the same as for the previously isolated compounds, on the basis of biogenetic considerations. This point needs to be better discussed and clarified in the main text.

Answer: From our previous experiments, compounds 2 and 4 were used same solvent in methanol-d4 for NMR test. By carefully comparing the NMR data, 2 and 4 were most alike except the glucose moiety. The 1H and 13C NMR spectrums were enclosed in this answer sheet. (Figure 14) Therefore, we considered the absolute configurations of compound 2 were identical as compound 4. For compounds 1 and 3, we applied with many bioassay experiments to find a suitable bioassay model and method. The remaining compounds were not enough for further tests.

-Absolute configuration of sugar moieties: As explained above, by NMR (J coupling analysis and 13C shift) only the relative configuration of the sugar can be inferred. The assignment of D and L sugar can only be done by X-ray, or after hydrolysis of the sugar moiety followed by derivation to a diastereoisomer and comparison with the corresponding D and L standards by GC or HPLC. As above, by only NMR data, the D and L absolute stereochemistry of sugar can only be hypothesized based on the same biogenetic pathway of the compounds previously isolated for which abs configuration has been determined.

Answer: Thank you for your advice. Our previous study of this plant led to the isolation of many glycoside compounds. [Biochemical Systematics and Ecology 2019, 86, 103905 and Molecules 2019, 24, 489], Therefore, the sugar moieties of compounds 1, 2, and 3 were hypothesized based on the same biogenetic pathway of the compounds previously isolated for which abs configuration has been determined.

Reviewer 2 Report

This paper is well written and it is structured logically, there are minor issues with grammar in a few sentences which should be corrected. The chemical and biological experiments are described in good detail.

The NMR analysis of the novel compounds is thorough, in Figure 2 it would be useful for the authors to provide the numbering around the glucosyl moiety, this would relate back to the signals with a double dash in Table 1.

The structure of Crocatin A has been determined, pipcroside A (1) was isolated as a white amorphous powder, were any attempts made to try and crystallize this material in order to get an x-ray structure? 

Possible biosynthetic pathways of the compounds are presented in the supplementary section, but there is no mention of this in the discussion. Given the structural differences in the bicyclic ring of compounds 1,2 and 4 compared with compound 3, it would be useful to include a discussion about this in the text.

Author Response

Answers of Reviewer 2' Comments

Manuscript ID: plants-1365581
Title: Unusual Bicyclo [3.2.1] Octanoid Neolignans from Leaves of Piper crocatum and Their Effect on Pyruvate Dehydrogenase Activity

Corresponding Author: Dr. Wei Li

We greatly appreciate your kind consideration on our manuscript.

According to reviewers’ comments, several parts of the manuscript have been corrected.

All comments were answered one by one and highlighted by yellow colour.

Reviewer 2' comments to Author:

This paper is well written, and it is structured logically, there are minor issues with grammar in a few sentences which should be corrected. The chemical and biological experiments are described in good detail.

The NMR analysis of the novel compounds is thorough, in Figure 2 it would be useful for the authors to provide the numbering around the glucosyl moiety, this would relate back to the signals with a double dash in Table 1.

Answer: This comment was corrected in the revised manuscript.

The structure of Crocatin A has been determined, pipcroside A (1) was isolated as a white amorphous powder, were any attempts made to try and crystallize this material in order to get an x-ray structure? 

Answer: Pipcroside A was formed as amorphous powder, even we tired many methods we could not get a good result from X-ray diffraction.

Possible biosynthetic pathways of the compounds are presented in the supplementary section, but there is no mention of this in the discussion. Given the structural differences in the bicyclic ring of compounds 1,2 and 4 compared with compound 3, it would be useful to include a discussion about this in the text.

Answer: Previously many bicyclo[3.2.1]octanoid neolignans were discovered from the Lauraceae plants. However, the biosynthetic pathways of this type of neolignans were rarely studied, but many of these compounds were isolated from various of plants. Given by the limited of literatures for this study, we could only refer with chemical total synthesis. We considered that without the explanation of biosynthetic pathways, it would be difficult to explain the differences in the bicyclic rings for this type of neolignan compounds. Therefore, we just mentioned in the text for this part.